# PluRel-to-RDB-PFN: Schema-Guided Synthetic Relational Pretraining

**Mohammad Sadeq Abolhasani** [1]   **Viswanath Ganapathy** [1]

**Abstract.** Relational Foundation Models (RFMs) require large-scale synthetic relational databases for pretraining, but existing approaches tightly couple data generation with the model training pipeline. We study whether PluRel, a general-purpose synthetic relational database generator, can serve as an external data source for RDB-PFN, a relational in-context learner originally pretrained with a 600K-task single-table warm-up followed by a $\sim$1.8M-task adaptation stage.

We build a conversion pipeline that maps PluRel-generated databases—including externally constructed binary prediction tasks—into the RDB-PFN training format and evaluate three curriculum strategies: SCHEMA-GUIDED FIRST (real-world schema then fully synthetic), FULLY SYNTHETIC (diverse synthetic schemas throughout), and SCHEMA-GUIDED LAST (fully synthetic then real-world schema). Using only $\sim$5,500 relational databases ($\sim$33K tasks)—roughly $55\times$ fewer tasks than the original protocol—and no single-table warm-up, our best curriculum (SCHEMA-GUIDED FIRST) achieves 0.6346 average ROC-AUC across 19 real benchmark tasks at 1024-shot context, recovering 87.6% of the published RDB-PFN performance (0.7245). At 64-shot context, the gap narrows to 93.8% (0.6116 vs. 0.6517). Our results demonstrate that external synthetic generators can provide useful pretraining signals for RFMs when combined with appropriate curriculum design and that exposure to a real-world schema early in training is substantially more effective than late-stage schema adaptation.

## 1. Introduction

Relational databases are the dominant data substrate in modern enterprises, yet building foundation models for relational prediction remains difficult. The natural pretraining corpus—large, diverse, real-world databases—is rarely available due to privacy, sensitivity, and schema heterogeneity. Recent work on Relational Foundation Models (RFMs) has therefore turned to synthetic data generation to bypass this bottleneck.

RDB-PFN (Wang et al., 2026) demonstrates that a lightweight in-context learner can be trained entirely on synthetic relational tasks generated from a structured prior. Its pipeline generates databases using a learned relational prior

(LayerDAG schemas, selective SCMs, bidirectional GNN content), linearizes them via Deep Feature Synthesis (DFS), and trains a Transformer to perform relational binary classification without gradient updates at test time. Separately, PluRel (Kothapalli et al., 2026) proposes a general-purpose framework for generating synthetic multi-table databases by sampling schemas via graph models, foreign-key connectivity via hierarchical stochastic block models, and row-level features via structural causal models. A key practical distinction is that RDB-PFN's generator jointly produces databases, features, *and* prediction tasks, but does not accept user-specified schemas. PluRel generates only databases—task construction must be added externally—but it can generate data under any user-provided SQL schema.

This paper studies whether these two systems can be connected. We ask two practical questions:

> *(1) Can PluRel-generated relational databases, converted into the RDB-PFN training format, provide useful pretraining signals for relational in-context learning?*
>
> *(2) Does the schema flexibility of PluRel — in particular, the ability to generate data under a fixed real-world schema — reduce the need for the massive pretraining corpus used in the original RDB-PFN protocol?*

To make this possible, we build a conversion pipeline that maps PluRel-generated databases into the RDB-PFN preprocessing stack. Since PluRel generates data but not prediction tasks, a critical component of our pipeline is the external construction of binary classification tasks from each synthetic database. We then apply the official RDB-PFN DFS preprocessing and train the RDB-PFN architecture on the resulting relational tasks.

Our experiments investigate three curriculum strategies that differ in whether and where a fixed real-world schema (derived from an enterprise system) is introduced during training. This design allows us to ask: *Does exposure to a real-world schema help at all? Is it more effective at the beginning or at the end of training? Or is a fully synthetic curriculum sufficient?* We also train the RDB-PFN architecture independently on each data pool (standalone pretraining) to isolate the contribution of individual curriculum stages.

The scale difference is substantial: the original RDB-PFN protocol trains on $\sim$600K single-table tasks followed by

[1]SAP Labs, LLC., Palo Alto 94304, United States. Correspondence to: Viswanath Ganapathy <viswa.ganapathy@sap.com>, Mohammad Sadeq Abolhasani <Rastin.Abolhasani@sap.com>.

*Proceedings of the $2^{nd}$ ICML on Foundation Models for Structured Data*, Seoul, South Korea. 2026. Copyright 2026 by the author(s).

~1.2M relational tasks (~1.8M total).

Our experiments use only ~5,500 relational databases producing ~33K tasks—roughly $55\times$ fewer—with no single-table warm-up. Despite this, our best curriculum recovers 87.6% of the published performance at 1024-shot context and 93.8% at 64-shot context, suggesting that schema-guided synthetic generation can partially compensate for scale.

**Contributions.**

1. **A PluRel-to-RDB-PFN pretraining pipeline** that converts PluRel-generated relational databases into the RDB-PFN training format, including external binary task construction, DFS linearization, and HDF5 assembly.

2. **A curriculum study on schema realism and synthetic diversity,** comparing three strategies (Schema-Guided First, Fully Synthetic, Schema-Guided Last) and four standalone baselines. Our results show that beginning with a real-world schema anchor and then progressing to diverse synthetic schemas is substantially more effective than the reverse order or fully synthetic training alone.

3. **Empirical evaluation on 19 real-world relational tasks** demonstrates that ~$55\times$ less training data can recover 87.6–93.8% of the published RDB-PFN performance when curriculum order and schema realism are appropriately designed.

## 2. Background

**Relational foundation models and synthetic pretraining.** Recent work extends the synthetic-data paradigm of tabular foundation models such as TabPFN (Hollmann et al., 2022) to relational databases. RDB-PFN (Wang et al., 2026) frames relational prediction as in-context learning: each synthetic RDB is linearized with DFS into a fixed-width table, and a lightweight bidirectional Transformer with column-wise and row-wise attention predicts query labels from labeled context rows without gradient updates. Its full protocol is substantially larger than ours: a 600K-task single-table warm-up followed by a Stage-2 corpus of roughly 1.8M tasks, including about 600K additional single-table tasks and 1.2M relational tasks. Our work keeps the RDB-PFN architecture and preprocessing pipeline, but replaces its native relational generator with PluRel-generated RDBs and removes the single-table warm-up.

**Synthetic relational database generation.** PluRel (Kothapalli et al., 2026) generates synthetic RDBs from scratch through schema sampling, primary–foreign key connectivity, and SCM-based feature generation. Schemas are sampled from random graph families such as Barabási–Albert, Watts–Strogatz, and reverse random trees; foreign keys are generated with hierarchical stochastic block models; and row features are produced by structural causal mechanisms conditioned on parent-table information. Unlike RDB-PFN's native generator, PluRel can also generate data under a fixed user-specified schema, making it useful for schema-guided pretraining. However, PluRel generates databases rather than prediction tasks, so our pipeline adds binary task construction before DFS preprocessing.

**Relational benchmarks and alternative generators.** RelBench (Robinson et al., 2024) provides standardized real-world relational prediction tasks, while graph-native RFMs such as griffin (Wang et al., 2025) operate directly on heterogeneous relational graphs. Other synthetic multi-table generators include diffusion-based methods such as ClavaDDPM (Pang et al., 2024) and RelDiff (Hudovernik et al., 2025), and graph-oriented synthetic pretraining approaches such as GraphPFN (Eremeev et al., 2025). Our study is complementary: rather than proposing a new model architecture, we test whether an external synthetic RDB generator can be plugged into an existing RDB-PFN-style relational in-context learning pipeline.

## 3. Method

### 3.1. PluRel-to-RDB-PFN Pipeline

Our pipeline converts PluRel-generated databases into RDB-PFN-compatible training data through five stages:

**(1) Database generation.** We use PluRel's `SyntheticDataset` API under two modes. In *schema-guided* mode, we provide a fixed SQL schema derived from a real-world enterprise ERP system (RelBench's rel-salt dataset: 4 tables, 31 columns, 6 FK edges), and PluRel generates synthetic row-level data under this schema. In *fully synthetic* mode, PluRel samples both schemas and data with configurable parameters. We use three fully synthetic configurations: *Small1* (2–4 tables, 1-hop DFS), *Small2* (2 to 4 tables, 2-hop DFS), and *Large* (5–8 tables, 1-hop DFS), with 5 to 15 columns, 50 to 200 entity rows, and 150 to 450 activity rows per table. We encode all columns as INTEGER/FLOAT (PluRel's type constraint), patch the SCM propagation to handle zero-FK-connection edge cases, and apply rank-based normalization to restore realistic temporal distributions.

**(2) Binary task construction.** For each database, we select up to 6 candidate target columns, preferring columns that are naturally binary. Otherwise, we binarize suitable categorical or numeric columns via median splits. Each task defines a target table, target column, and entity identifier.

**(3) Relational export.** Each database–task pair is exported in the DBInfer format expected by RDB-PFN's preprocessing stack.

**(4) DFS linearization.** We apply the official RDB-PFN DFS pipeline to produce fixed-width tabular representations

*Table 1.* Pretraining data comparison. Our PluRel-based pipeline uses ∼55× fewer tasks and no single-table warm-up compared to the original RDB-PFN protocol.

|  | RDB-PFN | Ours (PluRel) |
|---|---|---|
| Single-table tasks | ∼600K | 0 |
| Relational databases | ∼200K | ∼5,500 |
| Relational tasks | ∼1.2M | ∼33K |
| **Total tasks** | **∼1.8M** | **∼33K** |
| Schema control | No | Yes |

encoding multi-hop relational features.

**(5) Training data assembly.** The DFS-processed tasks are merged into HDF5 training files compatible with the RDB-PFN training loop.

### 3.2. Curriculum Strategies

We define four database pools, listed with their sizes:

- **Schema-Guided (SG)**: 500 databases were generated under the fixed real-world ERP schema.
- **Small1**: 3,000 fully synthetic databases (2–4 tables, 1-hop DFS).
- **Small2**: 1,000 fully synthetic databases (2–4 tables, 2-hop DFS).
- **Large**: 1,000 fully synthetic databases (5–8 tables, 1-hop DFS).

The total corpus is ∼5,500 databases with up to 6 tasks each, yielding ∼33K tasks. For comparison, the original RDB-PFN protocol uses ∼1.8M tasks (55× larger). Table 1 summarizes this comparison.

We evaluate three multi-stage curricula, training each stage sequentially:

**SCHEMA-GUIDED FIRST (SGF):** SG → Small1 → Small2 → Large. The model first learns from a fixed real-world schema with PluRel-generated content, and then encounters progressively more diverse synthetic schemas.

**FULLY SYNTHETIC (FS):** Small1 → Small2 (→ Large). Both schemas and data are fully synthetic throughout.

**SCHEMA-GUIDED LAST (SGL):** Small1 → Small2 → Large → SG. The model trains on diverse synthetic data first, then adapts to the fixed real-world schema.

Additionally, we train the RDB-PFN architecture independently on each data pool as **standalone baselines** (SG only, Small1 only, Small2 only, Large only) to isolate the contribution of individual stages.

## 4. Experiments

### 4.1. Setup

**Architecture and training.** All models use the RDB-PFN Transformer backbone (6 layers, 128 dims, 4 heads, 2.6M parameters) with Schedule-Free AdamW (learning rate $5 \times 10^{-4}$), fixed context of 600 rows × 30 columns. Each curriculum stage trains for 20K–50K steps.

*Table 2.* Average ROC-AUC across 19 binary classification tasks. **Bold**: best non-reference result. [†]Published RDB-PFN results using ∼55× more training data including 600K single-table warm-up tasks.

| Configuration | 64-shot | 1024-shot |
|---|---|---|
| *Reference (original protocol, ∼1.8M tasks)*[†] | | |
| RDB-PFN | 0.6517 | 0.7245 |
| *Curricula (this work, ∼33K tasks, relational only)* | | |
| SCHEMA-GUIDED FIRST | **0.6116** | **0.6346** |
| FULLY SYNTHETIC | 0.6028 | 0.6102 |
| SCHEMA-GUIDED LAST | 0.5544 | 0.5729 |
| *Standalone pretraining (this work)* | | |
| Small2 (2-hop) only | 0.5572 | 0.5772 |
| Schema-Guided only | 0.5431 | 0.5571 |
| Small1 (1-hop) only | 0.5146 | 0.5275 |
| Large only | 0.5112 | 0.5258 |

**Evaluation.** We evaluate 19 binary classification tasks from RelBench (Robinson et al., 2024) and DBInfer (Wang et al., 2024), spanning e-commerce (rel-amazon, rel-avito, Amazon, RetailRocket), social networks (rel-stack, StackExchange), sports (rel-f1), fashion (rel-hm), clinical trials (rel-trial), events (rel-event), and advertising (Diginetica, Outbrain, AVS). All tasks use ROC-AUC, evaluated at context sizes 64 and 1024 with 10 random seeds.

### 4.2. Results

**Main results.** Table 2 summarizes the average ROC-AUC across all 19 tasks. The SCHEMA-GUIDED FIRST curriculum achieves the strongest results: 0.6346 at 1024-shot and 0.6116 at 64-shot. FULLY SYNTHETIC is competitive at 64-shot (0.6028) but falls behind at 1024-shot (0.6102). SCHEMA-GUIDED LAST performs substantially worse (0.5544/0.5729), suggesting that introducing a fixed real-world schema *after* diverse synthetic training does not help and may overwrite previously learned patterns.

**Standalone pretraining is insufficient.** No single data pool exceeds an average ROC-AUC of 0.58 (Table 2, bottom). Small2 (2-hop DFS) is the strongest standalone pool (0.5772 at 1024-shot), likely because 2-hop features expose richer relational dependencies. Large databases alone perform the worst (0.5258), suggesting that complex schemas are too noisy without simpler relational stages as a foundation.

**Curriculum order matters substantially.** SCHEMA-GUIDED FIRST outperforms FULLY SYNTHETIC by 2-3 points and SCHEMA-GUIDED LAST by 5-6 points. Starting from a fixed real-world schema appears to provide a stable relational anchor—consistent table counts, stable FK patterns, and realistic column semantics—that enables the model to subsequently benefit from diverse synthetic schemas. Introducing the real-world schema *after* fully synthetic training (SCHEMA-GUIDED LAST) yields no im-

*Table 3.* Per-task ROC-AUC at 1024-shot context. SGF = SCHEMA-GUIDED FIRST (this work). [†]Published RDB-PFN. Tasks where SGF matches or exceeds RDB-PFN are marked.

| Task | SGF | RDB-PFN[†] |
|---|---|---|
| Amazon/churn | 0.4645 | 0.7179 |
| AVS/repeater | 0.5121 | 0.5599 |
| Diginetica/CTR | 0.5218 | 0.7004 |
| Outbrain/CTR | 0.5067 | 0.5352 |
| rel-amazon/item-churn | 0.7055 | 0.7821 |
| rel-amazon/user-churn | 0.5460 | 0.6479 |
| rel-avito/user-clicks | 0.5588 | 0.6266 |
| rel-avito/user-visits | 0.5350 | 0.6546 |
| rel-event/user-ignore | 0.6627 | 0.8273 |
| rel-event/user-repeat | 0.6549 | 0.7533 |
| rel-f1/driver-dnf | 0.6748 | 0.7188 |
| rel-f1/driver-top3 | 0.7488 | 0.8115 |
| rel-hm/user-churn | 0.5568 | 0.6648 |
| rel-stack/user-badge | 0.8005 | 0.8126 |
| rel-stack/engagement | 0.7249 | 0.8655 |
| rel-trial/study-outcome | 0.5075 | 0.6159 |
| RetailRocket/CVR | 0.7420 | 0.7708 |
| StackExchange/churn | 0.7803 | 0.8477 |
| StackExchange/upvote | **0.8535** | 0.8527 |
| **Average** | **0.6346** | **0.7245** |

provement over the synthetic-only curriculum, suggesting that late-stage schema adaptation is ineffective or causes catastrophic forgetting.

**Context-size analysis.** Our results strongly support the observation that context size changes the interpretation of performance. At 64-shot, the gap between SCHEMA-GUIDED FIRST and the published RDB-PFN is only $-0.0401$ (93.8% recovery). At 1024-shot, the gap widens to $-0.0899$ (87.6% recovery). This suggests that PluRel-based pretraining captures short-context relational patterns effectively, while the original RDB-PFN generator—with its bidirectional GNN content completion and $55\times$ larger corpus—provides stronger long-context structural generalization.

**Per-task analysis.** Table 3 shows per-task results for the best curriculum (SCHEMA-GUIDED FIRST) at 1024-shot alongside published RDB-PFN references.
Our model achieves 0.8535 on StackExchange/upvote (exceeding RDB-PFN's 0.8527) and is competitive on RetailRocket/CVR (0.7420 vs. 0.7708), rel-stack/user-badge (0.8005 vs. 0.8126), and rel-f1/driver-top3 (0.7488 vs. 0.8115). The largest gaps appear on tasks requiring fine-grained relational reasoning: Amazon/churn ($-0.25$), Diginetica/CTR ($-0.18$), and rel-event/user-ignore ($-0.16$).

## 5. Discussion

**Why does Schema-Guided First work best?** We hypothesize that a fixed real-world schema provides a structured relational inductive bias early in training: stable table counts, consistent FK patterns, and realistic column semantics. This

mirrors findings in curriculum learning (Bengio et al., 2009), where starting with cleaner, more structured examples accelerates convergence and improves final performance. The subsequent transition to diverse synthetic schemas then broadens the model's generalization without destabilizing the relational reasoning acquired in the first stage.

**What does the data-scale gap reveal?** The fact that $\sim$33K PluRel tasks recover 87.6–93.8% of the performance achieved by $\sim$1.8M RDB-PFN tasks suggests that the schema flexibility of PluRel—specifically, the ability to inject a real-world schema—partially compensates for the much smaller corpus. However, the widening gap at longer contexts (1024 vs. 64 shots) indicates that the original RDB-PFN generator, with its learned GNN content completion and far larger training set, produces richer inter-table statistical dependencies that become informative when many labeled examples are available in context.

**Limitations.** Our comparison is not scale-matched: the original RDB-PFN uses $\sim$55$\times$ more training data, including a single-table warm-up stage. PluRel's restriction to INTEGER/FLOAT types means categorical and text features are approximated, potentially reducing downstream task fidelity. Our task construction is external and heuristic; improving task generation may yield further gains. Finally, we evaluate only binary classification; extending to regression and multi-class tasks is future work.

**Future directions.** Scaling PluRel generation to match the original RDB-PFN data budget would clarify whether the remaining gap is due to generator quality or data scale. Combining PluRel's schema flexibility with RDB-PFN's content generation (e.g., using PluRel for schema/FK stages and a GNN for feature completion) is a promising hybrid direction. Extending schema-guided generation to multiple real-world schemas (e.g., from other RelBench datasets) could further improve curriculum diversity.

## 6. Conclusion

We have shown that PluRel, a general-purpose synthetic relational database generator, can serve as an external data source for RDB-PFN-style relational in-context learning. Using only $\sim$33K relational tasks—55$\times$ fewer than the original protocol—and no single-table warm-up, our best curriculum recovers 87.6–93.8% of the published RDB-PFN performance across 19 real-world tasks. The central finding is that *curriculum ordering and schema realism matter*: beginning with a real-world schema anchor and progressively introducing synthetic diversity substantially outperforms the reverse order and fully synthetic alternatives. These results suggest that the data generation and model training components of relational foundation models can be productively decoupled, enabling schema-controlled, privacy-preserving synthetic pretraining for enterprise relational data.

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

# A. Generated Database Statistics

This appendix reports summary statistics for the PluRel-generated relational databases used in our experiments. The fully synthetic corpora are generated with both synthetic schemas and synthetic table contents. The schema-guided corpora use a fixed real-world-inspired schema, while all table contents are still generated by PluRel. The `D_Large1_1000` statistics appeared twice in the raw logs with identical values; we include them once.

The main long curricula use the first four corpora in 4, totaling 5,467 generated relational databases with 5.97M rows and 215K columns before task augmentation. Including the schema-guided pilot corpus, the full generated collection contains 5,477 databases with 6.02M rows and 215K columns.

# B. Extended Results

This appendix provides extended ROC-AUC results for standalone pretraining and the three curriculum families. SG denotes schema-guided PluReldata, where the schema is fixed to a real-world-inspired schema and table contents are generated by PluRel. S1, S2, and L1 denote Small1, Small2, and Large1 fully synthetic PluRelcorpora. When a run was repeated with a longer training schedule, we report the better average result and omit repeat identifiers for readability.

**B.1. Aggregate Results**
**B.2. Task Abbreviations**
**B.3. Task-Wise Results at 64-Shot Context**
**B.4. Task-Wise Results at 1024-Shot Context**
**B.5. Standalone Schema-Guided Pilot at 32-Shot**

The schema-guided standalone model was also evaluated at 32-shot context. Its average ROC-AUC was 0.5431. We do not use this row in the main 64-shot comparison because it has a different context size, but we retain it here for completeness.

*Table 4.* Global statistics of the generated relational database corpora before task augmentation and DFSpreprocessing. SG denotes the schema-guided corpus; S1, S2, and L1 denote Small1, Small2, and Large1 fully synthetic corpora.

| Corpus | Source folder | DBs | Total rows | Rows / DB | Total cols. | Cols. / DB | Rows / table |
|---|---|---|---|---|---|---|---|
| S1 fully synthetic | B_Small1_3000 | 3,000 | 1,549,994 | 516 | 95,389 | 31 | 180 |
| S2 fully synthetic | C_Small2_1000 | 1,000 | 509,955 | 509 | 31,525 | 31 | 177 |
| L1 fully synthetic | D_Large1_1000 | 967 | 1,022,406 | 1,057 | 72,644 | 75 | 162 |
| SG schema-guided | SHOULD_BE_Full | 500 | 2,884,327 | 5,768 | 15,500 | 31 | 1,442 |
| SG schema-guided pilot | SHOULD_BE_SALT10 | 10 | 57,230 | 5,723 | 310 | 31 | 1,430 |

*Table 5.* Detailed distributional statistics of the generated corpora. Values are reported as min–max; mean $\pm$ standard deviation.

| Corpus | Rows / DB | Cols. / DB | Tables / DB | Rows / table | Cols. / table |
|---|---|---|---|---|---|
| S1 fully synthetic | 205–1,169; 516 $\pm$ 174 | 13–64; 31 $\pm$ 11 | 2–4; 2.86 $\pm$ 0.82 | 50–400; 180 $\pm$ 102 | 4–20; 11 $\pm$ 3 |
| S2 fully synthetic | 208–1,057; 509 $\pm$ 175 | 13–64; 31 $\pm$ 11 | 2–4; 2.87 $\pm$ 0.82 | 50–400; 177 $\pm$ 102 | 4–20; 10 $\pm$ 3 |
| L1 fully synthetic | 421–2,103; 1,057 $\pm$ 303 | 42–114; 75 $\pm$ 15 | 5–8; 6.52 $\pm$ 1.12 | 50–400; 162 $\pm$ 96 | 5–22; 11 $\pm$ 3 |
| SG schema-guided | 3,865–7,722; 5,768 $\pm$ 903 | 31–31; 31 $\pm$ 0 | 4–4; 4.00 $\pm$ 0.00 | 500–5,000; 1,442 $\pm$ 1,278 | 2–13; 7 $\pm$ 5 |
| SG schema-guided pilot | 3,999–7,032; 5,723 $\pm$ 1,003 | 31–31; 31 $\pm$ 0 | 4–4; 4.00 $\pm$ 0.00 | 515–5,000; 1,430 $\pm$ 1,291 | 2–13; 7 $\pm$ 5 |

*Table 6.* Aggregate extended results across 19 real relational binary classification tasks. The gap column reports the difference from the published RDB-PFNreference at the same context size. The SG-only 32-shot row has no corresponding RDB-PFNreference and is included for completeness.

| Family | Training stage | Context | Steps | Avg. ROC-AUC | Gap to RDB-PFN |
|---|---|---|---|---|---|
| Schema-guided only | SG | 32 | 150,000 | 0.5431 | – |
| Schema-guided only | SG | 1024 | 150,000 | 0.5571 | -0.1674 |
| Standalone | S1 | 64 | 20,000 | 0.5146 | -0.1371 |
| Standalone | S1 | 1024 | 20,000 | 0.5275 | -0.1970 |
| Standalone | S2 | 64 | 20,000 | 0.5572 | -0.0945 |
| Standalone | S2 | 1024 | 20,000 | 0.5772 | -0.1473 |
| Standalone | L1 | 64 | 20,000 | 0.5112 | -0.1405 |
| Standalone | L1 | 1024 | 20,000 | 0.5258 | -0.1987 |
| SG $\rightarrow$ Fully Synthetic | SG $\rightarrow$ S1 | 64 | 50,000 | 0.6063 | -0.0454 |
| SG $\rightarrow$ Fully Synthetic | SG $\rightarrow$ S1 | 1024 | 50,000 | 0.6138 | -0.1107 |
| SG $\rightarrow$ Fully Synthetic | SG $\rightarrow$ S1 $\rightarrow$ S2 | 64 | 100,000 | 0.6071 | -0.0446 |
| SG $\rightarrow$ Fully Synthetic | SG $\rightarrow$ S1 $\rightarrow$ S2 | 1024 | 100,000 | 0.6341 | -0.0904 |
| SG $\rightarrow$ Fully Synthetic | SG $\rightarrow$ S1 $\rightarrow$ S2 $\rightarrow$ L1 | 64 | 100,000 | **0.6116** | **-0.0401** |
| SG $\rightarrow$ Fully Synthetic | SG $\rightarrow$ S1 $\rightarrow$ S2 $\rightarrow$ L1 | 1024 | 100,000 | **0.6346** | **-0.0899** |
| Fully Synthetic | S1 $\rightarrow$ S2 | 64 | 50,000 | 0.6028 | -0.0489 |
| Fully Synthetic | S1 $\rightarrow$ S2 | 1024 | 50,000 | 0.6102 | -0.1143 |
| Fully Synthetic | S1 $\rightarrow$ S2 $\rightarrow$ L1 | 64 | 100,000 | 0.5701 | -0.0816 |
| Fully Synthetic | S1 $\rightarrow$ S2 $\rightarrow$ L1 | 1024 | 100,000 | 0.5438 | -0.1807 |
| Fully Synthetic $\rightarrow$ SG | S1 $\rightarrow$ S2 $\rightarrow$ L1 $\rightarrow$ SG | 64 | 100,000 | 0.5544 | -0.0973 |
| Fully Synthetic $\rightarrow$ SG | S1 $\rightarrow$ S2 $\rightarrow$ L1 $\rightarrow$ SG | 1024 | 100,000 | 0.5729 | -0.1516 |
| Published RDB-PFNreference | Original RDB-PFNprotocol | 64 | – | 0.6517 | – |
| Published RDB-PFNreference | Original RDB-PFNprotocol | 1024 | – | 0.7245 | – |

*Table 7.* Task abbreviations used in the extended task-wise result tables.

| Abbrev. | Task | Abbrev. | Task |
|---|---|---|---|
| Amz-C | Amazon churn | AVS-R | AVS repeater |
| Dig-CTR | Diginetica click-through rate | Out-CTR | Outbrain click-through rate |
| RA-I | rel-amazon item churn | RA-U | rel-amazon user churn |
| RAv-C | rel-avito user clicks | RAv-V | rel-avito user visits |
| RE-I | rel-event user ignore | RE-R | rel-event user repeat |
| RF1-D | rel-f1 driver DNF | RF1-T3 | rel-f1 driver top-3 |
| RHM-C | rel-hm user churn | RS-B | rel-stack user badge |
| RS-E | rel-stack user engagement | RT-O | rel-trial study outcome |
| RR-CVR | RetailRocket conversion rate | SE-C | StackExchange churn |
| SE-U | StackExchange upvote | Avg | Average across all 19 tasks |

*Table 8.* Extended 64-shot task-wise ROC-AUC results.

| Method | Amz-C | AVS-R | Dig-CTR | Out-CTR | RA-I | RA-U | RAv-C | RAv-V | RE-I | RE-R | RF1-D | RF1-T3 | RHM-C | RS-B | RS-E | RT-O | RR-CVR | SE-C | SE-U | Avg |
|---|---|---|---|---|---|---|---|---|---|---|---|---|---|---|---|---|---|---|---|---|
| S1 | 0.5064 | 0.4980 | 0.5532 | 0.4863 | 0.5102 | 0.5190 | 0.5105 | 0.5005 | 0.4662 | 0.4997 | 0.4959 | 0.5711 | 0.5159 | 0.4750 | 0.4716 | 0.5152 | 0.5909 | 0.5551 | 0.5363 | 0.5146 |
| S2 | 0.4886 | 0.5057 | 0.5024 | 0.5017 | 0.5063 | 0.5109 | 0.5499 | 0.4797 | 0.5766 | 0.5223 | 0.6227 | 0.6995 | 0.5207 | 0.7220 | 0.5285 | 0.4779 | 0.5774 | 0.6276 | 0.6672 | 0.5572 |
| L1 | 0.4983 | 0.4971 | 0.4787 | 0.4939 | 0.4992 | 0.4887 | 0.4936 | 0.5074 | 0.4858 | 0.5066 | 0.5036 | 0.5687 | 0.5149 | 0.5020 | 0.4970 | 0.4983 | 0.5227 | 0.5691 | 0.5879 | 0.5112 |
| SG→S1 | 0.5382 | 0.5049 | 0.5580 | 0.5026 | 0.6212 | 0.5249 | 0.5693 | 0.4906 | 0.6692 | 0.4778 | 0.6850 | 0.8011 | 0.5539 | 0.7287 | 0.6766 | 0.5076 | 0.5934 | 0.6911 | 0.8256 | 0.6063 |
| SG→S1→S2 | 0.5635 | 0.5137 | 0.5629 | 0.5083 | 0.6458 | 0.5538 | 0.5547 | 0.4877 | 0.5917 | 0.5398 | 0.6490 | 0.8006 | 0.5734 | 0.6366 | 0.5946 | 0.5211 | 0.6818 | 0.7174 | 0.8392 | 0.6071 |
| SG→S1→S2→L1 | 0.5085 | 0.5073 | 0.5423 | 0.5173 | 0.6097 | 0.5421 | 0.5578 | 0.4962 | 0.6725 | 0.5643 | 0.6288 | 0.7709 | 0.5631 | 0.8090 | 0.6736 | 0.5169 | 0.6197 | 0.6865 | 0.8330 | **0.6116** |
| S1→S2 | 0.5495 | 0.5153 | 0.5385 | 0.5004 | 0.6211 | 0.5173 | 0.5343 | 0.5127 | 0.6694 | 0.5751 | 0.6697 | 0.7884 | 0.5595 | 0.7033 | 0.5918 | 0.4981 | 0.6406 | 0.6818 | 0.7867 | 0.6028 |
| S1→S2→L1 | 0.5279 | 0.4960 | 0.5308 | 0.4997 | 0.5585 | 0.5308 | 0.5134 | 0.5184 | 0.5444 | 0.5301 | 0.6711 | 0.7798 | 0.5513 | 0.5304 | 0.5591 | 0.4706 | 0.6055 | 0.6082 | 0.8068 | 0.5701 |
| S1→S2→L1→SG | 0.5538 | 0.5146 | 0.4833 | 0.4976 | 0.5561 | 0.5149 | 0.4854 | 0.4782 | 0.6015 | 0.4678 | 0.6918 | 0.7756 | 0.5864 | 0.4075 | 0.4856 | 0.5043 | 0.5613 | 0.5425 | 0.8246 | 0.5544 |
| Published RDB-PFN | 0.6286 | 0.5172 | 0.6028 | 0.5106 | 0.7006 | 0.5791 | 0.5674 | 0.5061 | 0.7342 | 0.6055 | 0.6932 | 0.7952 | 0.6073 | 0.7729 | 0.7599 | 0.5474 | 0.6503 | 0.7669 | 0.8367 | 0.6517 |

*Table 9.* Extended 1024-shot task-wise ROC-AUC results.

| Method | Amz-C | AVS-R | Dig-CTR | Out-CTR | RA-I | RA-U | RAv-C | RAv-V | RE-I | RE-R | RF1-D | RF1-T3 | RHM-C | RS-B | RS-E | RT-O | RR-CVR | SE-C | SE-U | Avg |
|---|---|---|---|---|---|---|---|---|---|---|---|---|---|---|---|---|---|---|---|---|
| SG | 0.5264 | 0.4909 | 0.5289 | 0.4915 | 0.5145 | 0.5035 | 0.4804 | 0.5703 | 0.4854 | 0.5373 | 0.5787 | 0.7747 | 0.4933 | 0.5029 | 0.5377 | 0.5100 | 0.6354 | 0.6304 | 0.7934 | 0.5571 |
| S1 | 0.4742 | 0.5149 | 0.5936 | 0.5062 | 0.5269 | 0.5241 | 0.5408 | 0.4717 | 0.5122 | 0.5137 | 0.5198 | 0.6504 | 0.5183 | 0.4276 | 0.4425 | 0.5264 | 0.5487 | 0.6136 | 0.5963 | 0.5275 |
| S2 | 0.5095 | 0.5070 | 0.5464 | 0.5009 | 0.5233 | 0.5155 | 0.5322 | 0.5200 | 0.5675 | 0.4860 | 0.6648 | 0.7674 | 0.5164 | 0.7619 | 0.6021 | 0.4715 | 0.5443 | 0.6636 | 0.7671 | 0.5772 |
| L1 | 0.5097 | 0.4955 | 0.5556 | 0.4943 | 0.5222 | 0.4812 | 0.5026 | 0.5034 | 0.5584 | 0.4983 | 0.4963 | 0.5851 | 0.5015 | 0.4496 | 0.5172 | 0.5116 | 0.5475 | 0.5906 | 0.6699 | 0.5258 |
| SG→S1 | 0.5333 | 0.4992 | 0.5769 | 0.4959 | 0.7029 | 0.5366 | 0.5507 | 0.5790 | 0.7369 | 0.4422 | 0.7023 | 0.8092 | 0.5065 | 0.5720 | 0.6286 | 0.5089 | 0.7445 | 0.7045 | 0.8327 | 0.6138 |
| SG→S1→S2 | 0.5975 | 0.5143 | 0.5960 | 0.4925 | 0.7395 | 0.6106 | 0.5938 | 0.5830 | 0.5603 | 0.5417 | 0.7135 | 0.7891 | 0.5909 | 0.5816 | 0.6338 | 0.5313 | 0.7602 | 0.7750 | 0.8432 | 0.6341 |
| SG→S1→S2→L1 | 0.4645 | 0.5121 | 0.5218 | 0.5067 | 0.7055 | 0.5460 | 0.5588 | 0.5350 | 0.6627 | 0.6549 | 0.6748 | 0.7488 | 0.5568 | 0.8005 | 0.7249 | 0.5075 | 0.7420 | 0.7803 | 0.8535 | **0.6346** |
| S1→S2 | 0.5733 | 0.5048 | 0.5185 | 0.4855 | 0.6190 | 0.5183 | 0.5156 | 0.5232 | 0.7424 | 0.5737 | 0.7695 | 0.7829 | 0.5559 | 0.7829 | 0.6487 | 0.5164 | 0.5895 | 0.7192 | 0.7505 | 0.6102 |
| S1→S2→L1 | 0.5148 | 0.5014 | 0.5530 | 0.4962 | 0.5128 | 0.4993 | 0.5366 | 0.4494 | 0.5592 | 0.5401 | 0.6150 | 0.7228 | 0.5301 | 0.5774 | 0.5546 | 0.4870 | 0.5611 | 0.5105 | 0.6111 | 0.5438 |
| S1→S2→L1→SG | 0.4928 | 0.5005 | 0.5478 | 0.4914 | 0.5108 | 0.5199 | 0.4778 | 0.5253 | 0.5044 | 0.5223 | 0.6407 | 0.7669 | 0.5168 | 0.6314 | 0.6210 | 0.5071 | 0.7071 | 0.6300 | 0.7704 | 0.5729 |
| Published RDB-PFN | 0.7179 | 0.5599 | 0.7004 | 0.5352 | 0.7821 | 0.6479 | 0.6266 | 0.6546 | 0.8273 | 0.7533 | 0.7188 | 0.8115 | 0.6648 | 0.8126 | 0.8655 | 0.6159 | 0.7708 | 0.8477 | 0.8527 | 0.7245 |

*Table 10.* Schema-guided standalone 32-shot ROC-AUC results.

| Method | Amz-C | AVS-R | Dig-CTR | Out-CTR | RA-I | RA-U | RAv-C | RAv-V | RE-I | RE-R | RF1-D | RF1-T3 | RHM-C | RS-B | RS-E | RT-O | RR-CVR | SE-C | SE-U | Avg |
|---|---|---|---|---|---|---|---|---|---|---|---|---|---|---|---|---|---|---|---|---|
| SG 32-shot | 0.4633 | 0.4950 | 0.5551 | 0.4955 | 0.5086 | 0.5103 | 0.5449 | 0.4734 | 0.4694 | 0.5184 | 0.4878 | 0.5979 | 0.4890 | 0.6045 | 0.4965 | 0.5201 | 0.6088 | 0.7159 | 0.7644 | 0.5431 |

