# OpenReview forum: "PluRel-to-RDB-PFN: Schema-Guided Synthetic Relational Pretraining"
_ICML.cc/2026/Workshop/FMSD — FMSD @ ICML 2026 Poster_

### Official Review · Reviewer_cr8e · 2026-05-18
**Coherent Pipeline Study with Schema-Guided Gains**

**Rating:** 6
**Confidence:** 4

**Review:**

## Summary

This paper studies whether PluRel-generated synthetic relational databases can be used as an external pretraining source for RDB-PFN-style relational in-context learning. The authors build a conversion pipeline from PluRel databases to the RDB-PFN training format, including binary task construction, DFS linearization, and HDF5 assembly. They compare curriculum strategies and find that starting with a schema-guided corpus followed by fully synthetic schemas performs best, recovering 87.6% of published RDB-PFN performance at 1024-shot and 93.8% at 64-shot, while using far fewer training tasks.

## Strengths

**The paper addresses a relevant practical question.**
Decoupling synthetic relational database generation from RFM pretraining is useful. If external generators like PluRel can be plugged into RDB-PFN-style pipelines, this could make relational pretraining more flexible and controllable.

**The pipeline contribution is clear.**
The paper describes a concrete PluRel-to-RDB-PFN conversion process: database generation, external binary task construction, DBInfer export, DFS preprocessing, and training-data assembly. This is a useful engineering bridge between two existing systems.

**The curriculum comparison is interesting.**
The comparison between Schema-Guided First, Fully Synthetic, and Schema-Guided Last is one of the strongest parts of the paper. The result that early exposure to a fixed real-world schema helps more than late schema adaptation is plausible and useful.

**The evaluation covers many downstream tasks.**
The authors evaluate on 19 real relational binary classification tasks across multiple domains, including e-commerce, social networks, sports, fashion, clinical trials, events, and advertising.

## Areas for Improvement

**The schema-guided setup may be benchmark-specific.**
The schema-guided corpus is generated from a single fixed schema derived from RelBench’s rel-salt dataset / a real-world enterprise ERP schema, while evaluation is on real relational benchmark tasks. It is unclear whether the benefit comes from a generally useful schema-guided curriculum or from the chosen schema being unusually aligned with the evaluation distribution. The paper should test multiple schema-guided sources, including schemas that are held out from or deliberately different from the evaluation domains, to show that the effect generalizes.

**The explanation for why Schema-Guided First works is not fully isolated.**
The paper hypothesizes that starting with a fixed real-world schema provides a stable relational anchor, but there is limited direct evidence for this mechanism. More diagnostics would help separate the effects of schema realism, table size, row count, feature distribution, task construction, and curriculum order.

**The method still underperforms RDB-PFN, especially at 1024-shot.**
At 1024-shot, the best curriculum achieves 0.6346 average ROC-AUC versus 0.7245 for published RDB-PFN. The gap is substantial, and Table 3 shows large per-task deficits on several tasks, including Amazon/churn, rel-event/user-ignore, rel-amazon/item-churn, and rel-stack/engagement. The paper should frame the result as a promising partial recovery and be cautious about implying that PluRel can substitute for the original RDB-PFN generator without further scaling or controls.

**The main comparison to RDB-PFN is not clean.**
The paper emphasizes recovering 87.6–93.8% of published RDB-PFN performance with 55× fewer tasks. However, the original RDB-PFN protocol differs in several ways: data scale, generator, single-table warm-up, relational task construction, and likely pretraining distribution. Because many factors change at once, it is hard to attribute the result specifically to schema-guided PluRel generation or curriculum design.

**The task construction procedure may strongly affect results.**
PluRel generates databases but not prediction tasks, so the authors create binary tasks externally by selecting naturally binary columns or median-splitting suitable columns. This is a major part of the pipeline, but it is heuristic. The paper should analyze how sensitive results are to the task construction strategy.

## Justification of Score

I would rate this paper a **6**. The question is relevant, the pipeline is coherent, and the empirical results suggest that PluRel-generated databases can provide useful pretraining signal for RDB-PFN-style relational in-context learning. The curriculum comparison is also interesting, with Schema-Guided First outperforming fully synthetic and Schema-Guided Last training. My main reservations are that the comparison to published RDB-PFN is not scale-matched, the schema-guided effect is demonstrated using only one fixed real-world schema, and the explanation for why the curriculum works is not fully isolated.

---

### Official Review · Reviewer_YeJm · 2026-05-22
**Useful Study of Schema-Guided Synthetic Pretraining**

**Rating:** 7
**Confidence:** 4

**Review:**

## Summary

This paper studies synthetic-data-based training for relational database foundation models. It compares different synthetic data sources/setups and proposes a schema-guided-first curriculum. The main finding is that this curriculum improves training convergence and achieves reasonable performance with a much smaller training corpus. This suggests a useful direction for making relational foundation model pretraining more efficient.

## Strengths

The paper focuses on an important question: how to analyze and use synthetic relational data for training RDB foundation models. This topic is interesting, especially given two different synthetic-data designs can help isolate and better understand the role of the data generation process.

The proposed schema-guided-first curriculum is interesting and useful. It gives a practical insight into how synthetic data may be organized to improve training convergence and model performance.

## Areas for Improvement

My main concern is the final performance. Although the proposed method recovers a non-trivial fraction of the reference model’s performance, there is still a large gap from the full model. This makes me wonder whether the newly trained model is competitive with other strong ML baselines. It is also unclear how the model would behave with continued pretraining. The current results suggest promise, but they do not show whether the gap would close with more data or training.

## Detailed Comments

Including stronger baseline comparisons could better contextualize the absolute performance of the trained model.

A scaling study would be helpful to show whether continued pretraining can close the gap to the full model.

It would be useful to analyze the generated datasets themselves, not only the final model performance. Statistics such as schema diversity, table sizes, relation density, target construction, and task difficulty could help explain why the schema-guided-first curriculum works.

## Justification of Score

I would rate this paper as a 7. It addresses an important workshop-relevant problem and provides a useful empirical insight about schema-guided synthetic-data curricula. Although there is a remaining performance gap and limited comparison to other methods, I think the work is sufficient for this workshop.

---

### Official Review · Reviewer_19aA · 2026-05-22
**Good paper but is alignment is missing with real data**

**Rating:** 7
**Confidence:** 3

**Review:**

This paper proposes that a synthetic relational database generator (PluRel) can replace most of the original training pipeline for relational foundation models (RDB-PFN), cutting data needs while keeping 90% of performance. The paper is very well written, and the experiments are solid. The proposal is interesting as well, and this would benefit the broader community.

However, the issue I have is aligning the synthetic dataset with real-world data rather than treating it as a data augmentation technique. Specifically, when they ask whether synthetic relational data can actually transfer into in-context learning, they evaluate only the end results but never whether an external auditor (human or LLM) can ensure that the data aligns with real data at all.